# Computational Modeling of Lymph Filtration and Absorption in the Lymph Node by Boundary Integral Equations

**Alexey Setukha** [1,2] and **Rufina Tretiakova** [1,3,*]

[1] Moscow Center of Fundamental and Applied Mathematics at INM RAS, 119333 Moscow, Russia
[2] Research Computing Center, Lomonosov Moscow State University, 119992 Moscow, Russia
[3] Marchuk Institute of Numerical Mathematics of the RAS, 119333 Moscow, Russia
**\*** Correspondence: r.tretiakova@inm.ras.ru

**Abstract:** We develop a numerical method for solving three-dimensional problems of fluid filtration and absorption in a piecewise homogeneous medium by means of boundary integral equations. This method is applied to a simulation of the lymph flow in a lymph node. The lymph node is considered as a piecewise homogeneous domain containing porous media. The lymph flow is described by Darcy's law. Taking into account the lymph absorption, we propose an integral representation for the velocity and pressure fields, where the lymph absorption imitates the lymph outflow from a lymph node through a system of capillaries. The original problem is reduced to a system of boundary integral equations, and a numerical algorithm for solving this system is provided. We simulate the lymph velocity and pressure as well as the total lymph flux. The method is verified by comparison with experimental data.

**Keywords:** boundary integral equations; potential theory; filtration flow; collocation method

## 1. Introduction

In this paper, we develop a mathematical model of the filtration flow of a liquid in a piecewise homogeneous porous medium and apply it to the simulation of lymph flow in a lymph node.

The classical model of filtration flow relies on Darcy's law, which is valid in a linear porous medium. The numerical methods for solving problems of natural convection in porous media go back to the work of Chan et al. [1], where the finite difference method (FDM) was applied. Hickox and Gartling [2,3] used the finite element method (FEM), and Prasad and Kulacki [4] applied the finite volume method (FVM). The finite element and finite volume methods have been widely developed for solving various problems, including those in complex inhomogeneous regions (see, for example, [5–8]).

The traditional application of the filtration flows models is to the problem of fluid flow in different soils. The typical examples are the water intake problem and the oil flow to oil wells. On the other hand, modeling the fluid convection in living organisms and, in particular, the modeling of the lymph flow has its own specifics. There are just a few works that have considered the lymph flow in the lymph node by means of direct numerical simulations. The fluid flow description by Darcy's law in inhomogeneous regions (hydraulic conductivity is not constant) is due to Rose et al. [9]. In Moore et al. [10], a filtration domain was piecewise homogeneous, and the filtration flow was viscous, described by the Darcy–Brinkman law. The absorption of lymph into the blood in both papers was characterized by Starling's law, which relates the divergence of the velocity field to the fluid pressure. In all these models, the FEM approach was used.

If the domain consists of several homogeneous subdomains, then the boundary integral equations method is preferable, and the main goal of this work is to apply this method to the lymph filtration problem. In fact, this method is effective if an integral representation for a solution over domain and subdomains boundaries exists. If the latter is true, then

the approach has a number of advantages over the finite difference and finite element methods applied directly in the spatial domain. First, the dimension of the problem being solved is actually reduced. Two-dimensional integral equations (two-dimensional grids) are used instead of three-dimensional grids. Moreover, the differential equations outside the boundary surfaces and conservation laws for the numerical solutions are fulfilled. Finally, the integral representations of unknown functions allow us to calculate derivatives and various functionals without a significant loss of accuracy compared to the accuracy of numerical approximation of the functions themselves.

We can single out a number of papers in which the boundary element method was applied to the modeling of flows in porous media governed by Darcy's law or the Darcy–Brinkman law [11]. The solution of a viscous fluid filtration problem governed by the Darcy–Brinkman law using the boundary elements method (BEM) was constructed in [12,13]. However, these papers only considered two-dimensional problems in a homogeneous domain with the boundary condition set only on the outer boundary. The method of fundamental solutions (MFS) was presented in [14–16] for two-dimensional problems. This method was similar to the boundary element method, except that the solution was a superposition of partial solutions, the functions of source points located outside the flow area. That allowed avoiding singularity in the boundary condition. In this case, the method of boundary integral equations was applied to flows in one homogeneous region.

The application of the boundary integral equations method to the filtration problems in piecewise homogeneous domains was developed in the works of Piven et al., where two-dimensional flows were studied. A systematic presentation of these results can be found in [17]. A three-dimensional model of the fluid filtration was developed by Lifanov et al. [18]. The authors considered the filtration flows in a piecewise homogeneous domain with different types of external and internal boundaries and with the conjugation conditions at the interfaces between media with different properties. However, in all these works the flows with viscosity and absorption were not considered. The latter is important for the correct modeling of lymph flows in a lymph node.

In this paper, we use a simplified lymph node model consisting of two homogeneous domains: the outer one is the subcapsular sinus, the inner one is the T-cell and medullary zones. The inner region is characterized by a significantly higher hydraulic resistance and the presence of blood vessels absorbing lymph [19–22]. In the lymph nodes, the lymphatic and circulatory systems are conjugated. The overall fluid balance in the lymph node is determined by the pressure field, permeability, and location of blood vessels, in particular, high endothelial venules [20,22]. The structure of the lymph node is shown schematically in Figure 1.

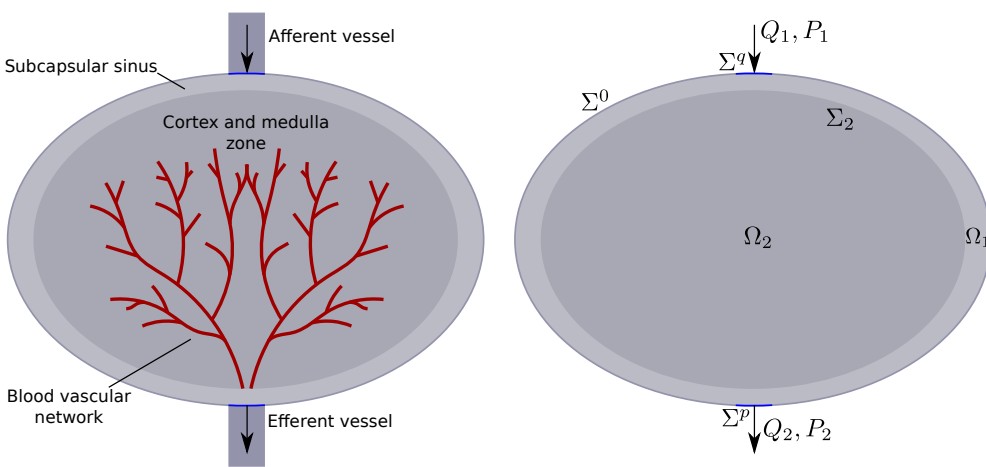

**Figure 1.** Simplified schematic representation of the lymph nodes: an external subdomain with low hydraulic resistance and an internal subdomain where lymph is absorbed into the blood vessels.

In the previous works of the authors [23,24], a solution to the problem of fluid filtration in a piecewise homogeneous porous medium governed by the Darcy–Brinkman law was developed. In [23], we implemented the reduction in the boundary value problem in partial derivatives to a system of boundary integral equations and also proved the equivalence of the boundary value problem and a system of boundary integral equations. In [24], we constructed and verified a numerical scheme for solving a system of boundary integral equations with piecewise constant approximations and collocations.

In this paper, we consider the problem of a stationary three-dimensional filtration flow with absorption in a piecewise homogeneous medium. The filtration is described by Darcy's law, and the absorption is described by Starling's law. A characteristic feature of such a flow is a nonzero divergence. We propose an integral representation for the velocity and pressure fields of the filtration flow. We also develop numerical schemes and verify the numerical results by comparison with experimental data of the filtration in the lymph node.

## 2. Mathematical Model

We denote $\Omega = \Omega_1 \bigcup \bar{\Omega}_2$ the filtration domain, bounded on the outside by a closed smooth surface $\Sigma_1$. $\Sigma_2$–boundary between the more permeable external subdomain $\Omega_1$ and the less permeable internal subdomain $\Omega_2$. An example of the computational domain is shown in Figure 1 (on the right).

Fluid filtration in both subdomains is described by Darcy's law for unknown fields of velocity $v$ and pressure $p$. In the external subdomain, the velocity satisfies the continuity equation. In the internal subdomain, Starling's law describes the absorption of fluid in capillaries. This model is similar to one given in paper [9]. Thus, the filtration–absorption problem can be written as a system of partial differential equations (1).

$$
\begin{cases}
v(x) = -\dfrac{\kappa_i}{\mu}\nabla p(x) & x \in \Omega_i,\ i = 1, 2, \\
\nabla \cdot v(x) = 0, & x \in \Omega_1, \\
\nabla \cdot v(x) = -L_b A(p(x) - p_b + \sigma \cdot \Delta\pi), & x \in \Omega_2;
\end{cases}
\tag{1}
$$

The model parameters are $\kappa_i$—the coefficients of hydraulic conductivity in subdomains $\Omega_i$, $i = 1, 2$, $\mu$—the lymph dynamic viscosity, $L_b$—the blood vessels' absorption coefficient, $A$—the surface area of the blood vessels, $p_b$—the blood pressure, $\Delta\pi$—the mean difference in the oncotic pressure of blood and lymph, and $\sigma$—the oncotic reflectance.

The outer boundary is divided into three parts: the openings of the afferent (inlet) vessels with the given lymph inflow ($\Sigma^q$), the openings of the efferent (output) vessels with the given pressure ($\Sigma^p$), and the impenetrable outer border with zero lymph flow ($\Sigma^0$). $\Sigma_1 = \Sigma^0 \bigcup \Sigma^q \bigcup \Sigma^p$. The inner boundary $\Sigma_2$ is homogeneous. The pressure and the normal component of the velocity vector must be continuous on this boundary.

We assume that each of the surfaces $\Sigma_m$, $m = 1, 2$ is oriented so that the outer side is considered positive. Let $n = n(x)$, $x \in \Sigma_m$, $m = 1, 2$, be the unit vector of the outer (positive) normal to the surface $\Sigma_m$ at the point $x$. The following boundary conditions are set on the boundaries of the domains:

$$
\begin{aligned}
n(x) \cdot v^-(x) - \xi(x) &= f_0(x), & x &\in \Sigma_1; & (2)\\
p^-(x) &= \psi(x), & x &\in \Sigma^p; & (3)\\
n(x) \cdot (v^+(x) - v^-(x)) &= 0, & x &\in \Sigma_2; & (4)\\
p^+(x) - p^-(x) &= 0, & x &\in \Sigma_2. & (5)
\end{aligned}
$$

Here, $f_0(x)$ is the known lymph flux, $f_0(x) \equiv 0$, $x \notin \Sigma^q$. $\psi(x)$ is the known pressure on $\Sigma^p$. An additional variable is introduced into the system of boundary conditions: $\xi(x)$ is the unknown lymph flux on the surface $\Sigma^p$, $\xi(x) \equiv 0$, $x \notin \Sigma^p$.

For simplicity, we introduce the notation:

$$\alpha_i = \frac{\mu}{\kappa_i}, \quad L = L_b A, \quad p_v = p_b - \sigma \cdot \Delta \pi$$

We exclude the variable $v$ from the system (1) by applying the $\nabla \cdot$ operator to the first equation of system (1) and substituting the second and third equations.

The pressure in the external subdomain $\Omega_1$ satisfies the Laplace equation.

$$\Delta p(x) = 0, \quad x \in \Omega_1. \tag{6}$$

In the internal subdomain $\Omega_2$, the pressure satisfies the Helmholtz equation:

$$\Delta p(x) - \lambda^2 p(x) = \lambda^2 p_v, \quad x \in \Omega_2, \quad \lambda = \sqrt{L\alpha_2}. \tag{7}$$

If $p = \tilde{p} + p_v$, then $\Delta \tilde{p} - \lambda^2 \, \tilde{p} = 0$.

*2.1. Simple and Double Layer Potentials*

The particular solutions of the Laplace and Helmholtz equations in the domain are the potentials of a simple and double layer on the boundary of the domain. The simple layer potential for the Helmholtz equation placed on the surface $\Sigma$ is the function

$$W_\lambda[\Sigma, h](x) = \int_\Sigma h(y) F_\lambda(x - y) d\sigma_y, \quad F(x - y) = \frac{\exp\{-\lambda|x - y|\}}{4\pi|x - y|}, \tag{8}$$

with $x \in \mathbb{R}^3 \setminus \Sigma$, and $h$ is the density of a simple layer potential, which is a function given on the surface $\Sigma$.

If $h \in C(\Sigma)$, then the simple layer potential $W[\Sigma, h](x)$ is defined by Formula (8) even for $x \in \Sigma$ (see [25]).

$$W_\lambda[\Sigma, h]^\pm(x) = W_\lambda[\Sigma, h](x), \quad x \in \Sigma. \tag{9}$$

Moreover, under certain smoothness conditions of the density $h(x)$, the following formula is valid for the boundary values of the gradient of the simple layer potential [25]:

$$\nabla W_\lambda[\Sigma, h]^\pm(x) = \nabla W_\lambda[\Sigma, h](x) \mp \frac{1}{2} h(x) \boldsymbol{n}(x) \quad x \in \Sigma, \tag{10}$$

where $\nabla W_\lambda[\Sigma, h](x)$ is the so-called direct value of the gradient of a simple layer potential, determined by the formula

$$\nabla W_\lambda[\Sigma, h](x) = \int_\Sigma h(y) \nabla_x F_\lambda(x - y) d\sigma_y, \quad x \in \Sigma,$$

where the integral is considered in the sense of a principal value.

The double layer potential for the Helmholtz equation $U_\lambda[\Sigma, g]$ with density $g$ set on the surface $\Sigma$ is the following integral operator:

$$U_\lambda[\Sigma, g](x) = \int_\Sigma g(y) \frac{\partial F_\lambda(x - y)}{\partial n_y} d\sigma_y, \quad x \in \mathbb{R}^3 \setminus \Sigma. \tag{11}$$

The double layer potential with density $g \in C(\Sigma)$, defined by Expression (11) at the points of the space $\mathbb{R}^3 \setminus \Sigma$, is a harmonic function on this set. The double layer potential can be extended on the surface $\Sigma$ from the external and internal domains $\Omega_1$ and $\Omega_2$ (see [25]). The boundary values of the function $U_\lambda[\Sigma, g]$ on the surface $\Sigma$ are given by the expression:

$$U_\lambda[\Sigma, g](x)^\pm = U_\lambda[\Sigma, g](x) \pm \frac{g(x)}{2} \quad x \in \Sigma, \tag{12}$$

with $U_\lambda[\Sigma, g](x)$ being the direct value of the double layer potential obtained directly from Expression (11) at point $x \in \Sigma$.

If the double layer potential density $g(x)$ satisfies a certain smoothness condition, then the vector field $\nabla U_\lambda[\Sigma, h]$ can be continued onto the surface $\Sigma$ from each of the domains $\Omega_1$ and $\Omega_2$. For the boundary values of the gradient of the function $U_\lambda[\Sigma, h]$, the formula is valid [25]:

$$\nabla U_\lambda[\Sigma, g]^{\pm}(x) = \nabla U_\lambda[\Sigma, g](x) \pm \frac{1}{2}\text{Grad } g(x), \quad x \in \Sigma. \tag{13}$$

The direct value of the double layer potential's gradient at the boundary of a closed surface is the following expression:

$$\nabla U_\lambda[\Sigma, g](x) = \int_\Sigma [\text{Grad } g(y) \times n(y)] \times \nabla_x F(x - y) d\sigma_y. \tag{14}$$

The simple or double layer potential for the Laplace equation is a particular case of the Helmholtz potential with $\lambda = 0$.

*2.2. Boundary Integral Equations*

Following the reasoning given in [23], we put a simple layer potential on the outer surface $\Sigma_1$ and a simple layer and double layer potentials on the inner boundary $\Sigma_2$. Thus, the field of velocities and pressures in the regions $\Omega_1$, $\Omega_2$ are sought in the following form:

$$v = \nabla W_{\lambda_i}[\Sigma_1, h] + \nabla U_{\lambda_i}[\Sigma_2, g] + \nabla W_{\lambda_i}[\Sigma_2, h], \tag{15}$$
$$p = -\alpha_i\left(W_{\lambda_i}[\Sigma_1, h] + U_{\lambda_i}[\Sigma_2, g] + W_{\lambda_i}[\Sigma_2, h]\right) + \chi_{\Omega_2} p_v. \tag{16}$$

Here, $\lambda_1 = 0$ in $\Omega_1$, $\lambda_2 = \sqrt{L\alpha_2}$ in $\Omega_2$, and $\chi_{\Omega_2}$ is the domain indicator function for $\Omega_2$. The advantage of this representation is that the velocity and pressures are expressed by the same operators in different domains.

The velocity and pressure in the integral representations (15) and (16) satisfy system (1). The integral representations (15) and (16) are substituted into the boundary conditions (2)–(5). We construct a system of integral equations for the unknown densities of the simple and double layer potentials $h$, $g$ in which each equation corresponds to one of the boundary conditions (2)–(5).

$$\frac{h}{2} + \boldsymbol{n} \cdot \nabla W_0[\Sigma_1, h] + \boldsymbol{n} \cdot \nabla U_0[\Sigma_2, h] + \boldsymbol{n} \cdot \nabla W_0[\Sigma_2, h] - \xi = f_0, \quad x \in \Sigma_1, \tag{17}$$

$$W_0[\Sigma_1, h] + U_0[\Sigma_2, h] + W_0[\Sigma_2, h] = -\frac{\psi}{\alpha_1}, \quad x \in \Sigma_1, \tag{18}$$

$$\alpha_1 W_0[\Sigma_1, h] - \alpha_2 W_{\lambda_2}[\Sigma_1, h] + \frac{\alpha_1 + \alpha_2}{2} g +$$
$$+ \alpha_1 U_0[\Sigma_2, h] - \alpha_2 U_{\lambda_2}[\Sigma_2, h] + \alpha_1 W_0[\Sigma_2, h] - \alpha_2 W_{\lambda_2}[\Sigma_2, h] = -p_v, \quad x \in \Sigma_2, \tag{19}$$

$$\boldsymbol{n} \cdot \left(\nabla W_0[\Sigma_1, h] - \nabla W_{\lambda_2}[\Sigma_1, h]\right) + \boldsymbol{n} \cdot \left(\nabla U_0[\Sigma_2, h] - \nabla U_{\lambda_2}[\Sigma_2, h]\right) -$$
$$- h + \boldsymbol{n} \cdot \left(\nabla W_0[\Sigma_2, h] - \nabla W_{\lambda_2}[\Sigma_2, h]\right) = 0, \quad x \in \Sigma_2, \tag{20}$$

The solution functions $h$, $g$ of the boundary integral equation system allow calculation of the velocity and pressure fields in integral form (15) and (16) in the filtration domain $\Omega$.

**3. Numerical Method**

We use the method of piecewise constant approximations and collocations for the numerical solution of the system of integral equations. This method has a low order of accuracy, but it also has such advantages as versatility, simplicity of implementation, and

reliability. This is why the method is convenient for conducting computational experiments to test new mathematical models.

The surfaces $\Sigma_m$, $m = 1, 2$, are approximated by a system of triangular and rectangular cells with all the cells' vertices lying on the surface being approximated.

$$\Sigma_1 \simeq \tilde{\Sigma}_1 = \{\sigma_k\}, \ k = 1, ..., N^1, \quad \Sigma_2 \simeq \tilde{\Sigma}_2 = \{\sigma_k\}, \ k = N^1 + 1, ..., N.$$

Each cell has its local basis consisting of a normal vector $n_k$ and two tangential vectors $\tau_k^1$, $\tau_k^2$. We also set a collocation point $x_k$ on each cell $\sigma_k$, $k = 1, ..., N$. The example of surface approximation is given in papers [24,26].

We approximate unknown functions $h, g$ by piecewise constant functions $\tilde{h}, \tilde{g}$ set on the approximate surfaces $\tilde{\Sigma}_m$, $m = 1, 2$. We set

$$\tilde{h}(x) = h_k, \ \tilde{g}(x) = g_k, \ x \in \sigma_k \setminus \partial\sigma_k, \tag{21}$$

$k = 1, ..., N$ for function $h$, and $k = N^1 + 1, ..., N$ for function $\tilde{g}$. These qualities are satisfied at the internal points of the cells. The $\partial\sigma_k$ is the cell edge.

### 3.1. Approximation of the Integral Operators

Let $S$ be one of the surfaces $\Sigma_n$, $n = 1, 2$, and $\tilde{S}$ is its approximation. We approximate the direct values of the integral operators $W_{\lambda_m}[\tilde{S}, \tilde{h}]$, $U_{\lambda_m}[\tilde{S}, \tilde{g}]$, $\nabla W_{\lambda_m}[\tilde{S}, \tilde{h}]$ in collocation points $x_i \in \Sigma_n$, $n = 1, 2$, and the following formulas:

$$W_{\lambda_m}[\tilde{S}, \tilde{h}](x_i) = \sum_{\sigma_k \in \tilde{S}} h_k W_{ik}^{\lambda_m}, \quad W_{ik}^{\lambda_m} = \frac{1}{4\pi} \int_{\sigma_k} \frac{e^{-\lambda_m r_i}}{r_i} d\sigma_y,$$

$$U_{\lambda_m}[\tilde{S}, \tilde{g}](x_i) = \sum_{\sigma_k \in \tilde{S}} g_k U_{ik}^{\lambda_m}, \quad U_{ik}^{\lambda_m} = \frac{1}{4\pi} \int_{\sigma_k} (x_i - y, n_k) \frac{e^{-\lambda_m r_i}(\lambda_m r_i + 1)}{r_i^3} d\sigma_y,$$

$$\nabla W_{\lambda_m}[\tilde{S}, \tilde{h}](x_i) = \sum_{\sigma_k \in \tilde{S}} h_k w_{ik}^{\lambda_m}, \quad w_{ik}^{\lambda_m} = \frac{-1}{4\pi} \int_{\sigma_k} (x_i - y) \frac{e^{-\lambda_m r_i}(\lambda_m r_i + 1)}{r_i^3} d\sigma_y.$$

Here, $r_i = |x_i - y|$, and $m$ is the number of subdomains where the direct value of the potentials are calculated. Note that if $i = k$, the integrals $W_{ik}^{\lambda_m}$ are improper absolutely convergent, and $U_{ik}^{\lambda_m}$, $w_{ik}^{\lambda_m}$ are singular considered as the principal value.

The direct values of the double layer potential gradient $\nabla U_\lambda[S, g]$ in collocation points $x_i \in \Sigma^m$ are approximated the following way:

$$\nabla U_{\lambda_m}[S, g] = \sum_{\sigma_k \in \tilde{S}} g_k u_{ik}^{\lambda_m}, \quad u_{ik}^{\lambda_m} = \nabla \int_{\sigma_k} \frac{\partial F_{\lambda_m}(x_i - y)}{\partial n_y}$$

To calulate the integral $u_{ik}^\lambda$, we separate the singularity and integrate it analytically. We decompose the function $F_\lambda(x - y)$ into

$$F_\lambda(x - y) = F_0(x - y) + \tilde{F}_\lambda(x - y), \quad F_0(r) = \frac{1}{4\pi r}, \quad \tilde{F}_\lambda(r) = \frac{e^{-\lambda r} - 1}{4\pi r}.$$

with $r = |x - y|$. So, the integral is decomposed the following way:

$$u_{ik}^\lambda = u_{ik}^0 + \tilde{u}_{ik}^\lambda, \quad u_{ik}^0 = \nabla \int_{\sigma_k} \frac{\partial F_0(x_i - y)}{\partial n_y} d\sigma_y, \quad \tilde{u}_{ik}^\lambda = \nabla \int_{\sigma_k} \frac{\partial \tilde{F}_\lambda(x_i - y)}{\partial n_y} d\sigma_y.$$

To caluclate $u_{ik}^0$, we use the Biot–Savart law [27,28], and integral $\tilde{u}_{ik}^\lambda$ is improper convergent.

The integrals in expressions $W_{ik}^\lambda$, $U_{ik}^\lambda$, $w_{ik}^\lambda$, and $\tilde{u}_{ik}^\lambda$ are calculated approximately by the rectangular method. Such a scheme for improper and singular integrals was described in the paper [24].

*3.2. Approximation of the Boundary Equations*

To calculate the approximate solution, we use the collocation method. Equations (17)–(20) must be fulfilled at collocation points $x_i$, $i = 1, ..., N$.

$$\frac{h_i}{2} + \sum_{k=1}^{N} h_k (\boldsymbol{n}_i \cdot \boldsymbol{w}_{ik}^0) + \sum_{k=N_1+1}^{N} g_j (\boldsymbol{n}_i, \boldsymbol{u}_{ik}^0) - \xi_i = f_{0,i}, \quad i = 1, ..., N_1, \tag{22}$$

$$\sum_{k=1}^{N} h_k W_{ik}^0 + \sum_{k=N_1+1}^{N} g_j U_{ik}^0 = -\frac{\psi_i}{\alpha_1}, \quad i = 1, ..., N_1, \tag{23}$$

$$\sum_{k=1}^{N} h_k (\alpha_1 W_{ik}^0 - \alpha_2 W_{ik}^{\lambda_2}) + \frac{\alpha_1 + \alpha_2}{2} g_i +$$
$$+ \sum_{k=N_1+1}^{N} g_k (\alpha_1 U_{ik}^0 - \alpha_2 U_{ik}^{\lambda_2}) = -p_v, \quad i = N_1 + 1, ..., N, \tag{24}$$

$$- h_i + \sum_{k=1}^{N} h_k \boldsymbol{n}_i \cdot (\boldsymbol{w}_{ik}^0 - \boldsymbol{w}_{ik}^{\lambda_2}) + \sum_{k=1}^{N} g_k (\boldsymbol{n}_i \cdot \tilde{\boldsymbol{u}}_{ik}^{\lambda_2}) = 0, \quad i = N_1 + 1, ..., N. \tag{25}$$

After solving the system of linear Equations (22)–(25), we calculate the velocity and pressure fields in point $x \in \Omega_m$, $m = 1, ..., N_d$ by the approximations of Formulas (15) and (16):

$$\boldsymbol{v} = \sum_{k=1}^{N} h_k \nabla W_{\lambda_m}[\sigma_k, e] + \sum_{k=N_1+1}^{N} g_k \nabla U_{\lambda_m}[\sigma_k, e], \tag{26}$$

$$p = \chi_{\Omega_2} p_v - \alpha_m \left( \sum_{k=1}^{N} h_k W_{\lambda_m}[\sigma_k, e] + \sum_{k=N_1+1}^{N} g_k U_{\lambda_m}[\sigma_k, e] \right), \tag{27}$$

with $e \equiv 1$. The integrals in the expressions $W_{\lambda_m}[\sigma_k, e]$, $U_{\lambda_m}[\sigma_k, e]$, $\nabla W_{\lambda_m}[\sigma_k, e]$, and $\nabla U_{\lambda_m}[\sigma_k, e]$ are proper in the inner points of the domains $\Omega_1$, $\Omega_2$. These integrals are also calculated by the rectangle method.

**4. Results**

The papers [29–31] contained the measurements of the lymph flow, pressure, and protein concentration in the popliteal lymph nodes of dogs. To verify the model, we used the experimental data given in these papers. We constructed a geometrical model approximating the lymph node and set the boundary conditions according to the experimental data. Then, we solved the parametrical optimization problem to fit the experiment.

The fluid exchange between the lymph node and the circulatory system is described by Starling's law (1), the third equation. The value of parameter $\sigma$ was set at 0.88, according to [10]. The experimental data presented in [31] contained the blood pressure $p_b$ and the protein concentration in the blood $C_b$ and the lymph $C_l$. The oncotic pressure was calculated by the formula $\Delta \pi = \frac{C_b - C_l}{M} R T$, with $M = 67.2$ kg/mol as the molar mass of proteins, $R = 8.31$ as the universal gas constant, and $T = 301$ K as the absolute temperature. Given these data, we calculated $p_v = p_b - \sigma \Delta \pi$.

The experiment was carried out on the lymph node with one inlet opening $\Sigma^q$ with a fixed flow $Q_1$ and one outlet opening $\Sigma^p$ with fixed pressure $P_2$ (see Figure 1 on the right). The external subdomain $\Omega_1$ was ellipsoid with radii $R_x = R_z = 0.35$ mm, and $R_y = 0.5$ mm. The internal subdomain $\Omega_2$ was ellipsoid with radii $R_x = R_z = 0.315$ mm, and $R_y = 0.45$ mm. The afferent and efferent lymphatic vessels were considered to be cylindrical with the radius $r = 0.075$ mm and the centers $z^q$ and $z^p$ located on the upper and bottom poles of $\Sigma_1$, respectively.

The surfaces $\Sigma^q$ and $\Sigma^p$ were approximated as follows:

$$\tilde{\Sigma}^q = \{\sigma_i : |x_i - z^q| < r\}, \qquad \tilde{\Sigma}^p = \{\sigma_i : |x_i - z^p| < r\}.$$

We approximated the velocity functions $f_0$ and $\xi$ on the parts of the boundary $\Sigma^q$ and $\Sigma^p$, so that the flow through the openings had a parabolic profile. The pressure on the outlet was assumed to be constant.

$$f_{0,i} = Q_1 \frac{r^2 - |x_i - z^q|^2}{\sum_{k:\sigma_k \in \tilde{\Sigma}^q}(r^2 - |x_k - z^q|^2)s_k}, \quad \psi = P_2. \tag{28}$$

The function $\xi$ on part of surface $\tilde{\Sigma}^p$ was approximated in a similar way.

The known experiment did not contain the complete description of all the parameters that arise in the mathematical formulation of the problem, since it is extremely difficult to calculate such measurements. Therefore we used the following approach: part of the experimental data were used as the initial parameters of model, and the other part of the data were used for validation of the model. We denoted the model parameters $\theta = \{L, \alpha_1, \alpha_2\}$. We approximated the boundary integral Equations (17)–(20) by a numerical scheme (22)–(25) and solved the corresponding linear system with parameters $\theta$ and boundary conditions (28) with the values $Q_1$ and $P_2$ taken from paper [31] (Table 1).

To compare the mathematical model prediction with the experimental data values of the outlet flow $Q_2$ and inlet pressure $P_1$, we calculated the pressure $\bar{P}_1$ on $\tilde{\Sigma}_1^q$ and the flow $\bar{Q}_2$ on $\tilde{\Sigma}_1^p$ using the following formulas:

$$\bar{Q}_2 = \sum_{\sigma_i \in \tilde{\Sigma}_1^p} n_i \cdot v_i^- \cdot s_i, \quad \bar{P}_1 = \frac{\sum_{\sigma_i \in \tilde{\Sigma}_1^q} p_i^- \cdot s_i}{\sum_{\sigma_i \in \tilde{\Sigma}_1^q} s_i}$$

The article [31] provided data on several animals (greyhound dogs) with a number of experimental values for each animal. Assuming that different animals corresponded to different values of the $\theta$ parameters, we calculated the optimal values of the parameters that minimized the functional $\Phi_{err}$ for all experiments for each animal.

$$\Phi_{err}(\theta) = \left( \sum_{i=1}^{N_{exp}} \frac{(Q_2^i - \bar{Q}_2^i(\theta))^2}{Q_{avg}} + \frac{(P_1^i - \bar{P}_1^i(\theta))^2}{P_{avg}} \right)^{\frac{1}{2}},$$

$$Q_{avg} = \sum_{i=1}^{N_{exp}} \frac{(Q_2^i)^2}{N_{exp}}, \quad P_{avg} = \sum_{i=1}^{N_{exp}} \frac{(P_1^i)^2}{N_{exp}}. \tag{29}$$

Here, $Q_2^i$ and $P_1^i$ were the $i$-th experiment data values, and $\bar{Q}_2^i$ and $\bar{P}_1^i$ were calculated by the numerical model for the $i$-th experiment boundary condition values. To solve the optimization problem, we used the Nelder–Mead method.

The values of the functional as well as the optimal values of the parameters are given in Table 1. In the experiment, the value of $P_2$ varied, while $p_v$ had a small variation, and $Q_1$ was constant. So, the comparison of the simulation results with the experimental data can be represented as the dependence of $P_1$ and $Q_2$ on $P_2$ (see Figures 2 and 3).

**Table 1.** Parameters providing optimal approximation to the experimental data.

| Dog | $\Phi_{err}$ | $L$ | $\alpha_1$ | $\alpha_2$ |
|-----|--------------|-----|------------|------------|
| 1 | 0.16 | 6.22 | $1.2 \cdot 10^{-3}$ | 3.08 |
| 2 | 0.26 | 8.42 | $0.7 \cdot 10^{-3}$ | 4.46 |
| 3 | 0.24 | 7.94 | $0.6 \cdot 10^{-3}$ | 3.67 |
| 4 | 0.19 | 6.17 | $0.7 \cdot 10^{-3}$ | 3.49 |

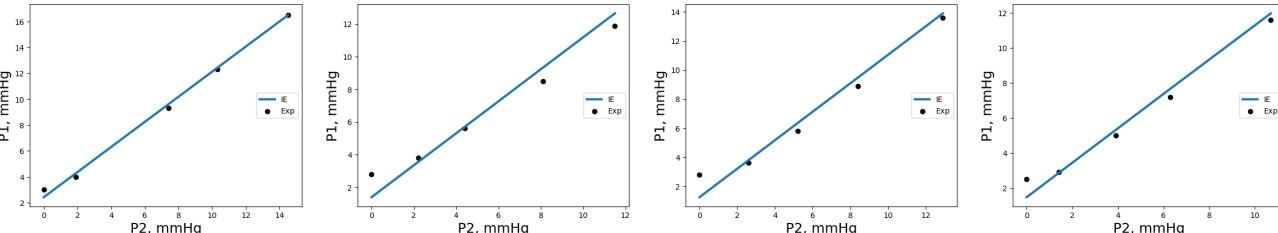

**Figure 2.** Pressure values $P_1$ at the inlet opening depending on pressure $P_2$ at the outlet opening. IE—integral equation model, Exp—experimental data.

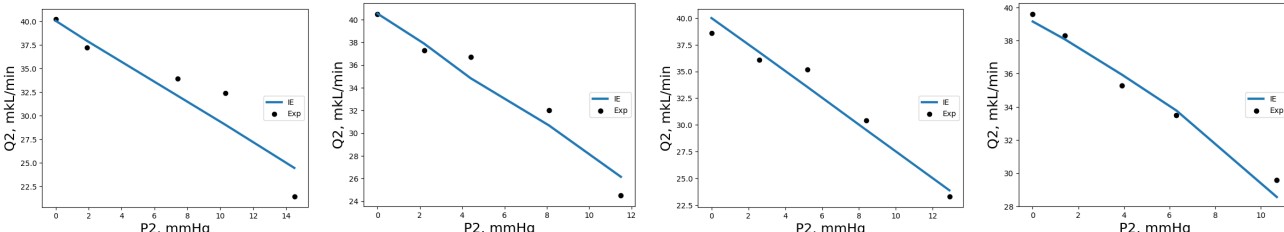

**Figure 3.** Flow values $Q_2$ at the outlet opening depending on pressure $P_2$ at the outlet opening. IE—integral equation model, Exp—experimental data.

The value of the deviation of the results from the experiment $\Phi_{err}$ demonstrates that the model accurately approximated the data for four animals.

## 5. Discussion

In this paper, we proposed a model of the fluid filtration flow in a piecewise homogeneous domain containing porous medium and applied it to the simulation of the lymph flow in the lymph node. We also assumed that the fluid was absorbed in the inner subdomain. A mixed boundary condition was set on the outer boundary of the domain (the pressure was set on the outlet opening, the flow was set on the inlet opening, and the impermeability condition was set on the remaining part of the boundary). At the interface between the media, the conjugation conditions were set (the continuity of the pressure and the normal component of the fluid velocity). An integral representation of the velocity and pressure fields in terms of the potentials of a simple layer and double layer at the domain boundary was proposed. The boundary value problem was reduced to a system of boundary integral equations for unknown densities of simple layer and double layer potentials.

A numerical solution scheme based on the method of piecewise constant approximations and collocations was constructed for the system of integral equations. The solution of the approximated system of boundary integral equations allowed calculation of the velocity and pressure fields in any inner point of the domain.

The model predictions were also compared with the existing experimental data on lymph filtration in the lymph node. An agreement between the values of the total flow characteristics obtained by numerical simulation and the experimental data was reached. The analysis of the obtained test results, in particular, showed the applicability of the developed numerical model for the simulation of the considered class of filtration flows.

The results of this work were used for the development of an artificial neural network model describing the lymph node drainage function, which was described in [32].

**Author Contributions:** Conceptualization, A.S. and R.T.; methodology, A.S.; software, R.T.; validation, R.T.; investigation, R.T.; writing—original draft preparation, R.T.; writing—review and editing, A.S.; visualization, R.T.; supervision, A.S. All authors have read and agreed to the published version of the manuscript.

**Funding:** This research was funded by the Moscow Center for Fundamental and Applied Mathematics at INM RAS (agreement with the Ministry of Education and Science of the Russian Federation No. 075-15-2019-1624) and partially funded by the Russian Science Foundation (grant number 18-11-00171).

**Institutional Review Board Statement:** Not applicable

**Informed Consent Statement:** Not applicable

**Data Availability Statement:** Data is contained within the article

**Conflicts of Interest:** The authors declare no conflicts of interest.

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
