# Peer review of "Computational Modeling of Lymph Filtration and Absorption in the Lymph Node by Boundary Integral Equations"

_algorithms, doi:10.3390/a15100388_

Round 1
Reviewer 1 Report
I think the topic of the paper is interesting. The BVP integral equations are a powerful tool in order to solve several differential problems.
Nevertheless I underline that some of the approximating tools are too much simple/elementary (for instance regarding the approximation of the singular integrals or the use of the rectangular quadrature). I understand that this is due to the high complexity of the model, but I fear that if a convergence analysis were conducted, the results would be rather poor in terms of order of convergence and speed of the method.
The comparison with experimental date assure the consistency of the numerical strategy at least in the considered cases. It would be interesting, in further investigation, to study the stability of the whole procedure.
Author Response
We thank the reviewer
We agree with the remark about the simplicity of the piecwise constant method.
But despite the low order of approximation this method is quite reliable, versatile and easy to implement.
That's why it is convenient to use this method to check the new mathematical model.
We added this explanation in the paper, section "Numerical Methods".
We also made a revision of the English language in the sections: abstract, introduction, discussion.
Reviewer 2 Report
The paper models the lymph filtration in a lymph node. The authors introduce the differential equations (Darcy law for the relation velocity/pressure gradient and the incompressibility equation) and the appropriate boundary conditions. The apply the potential theory to reduce this problem to a boundary integral equation. This equation is further solved by a numerical method which generalizes I.Lifanov's scheme. What is especially interesting is that the authors compare the numerical results with experimental data and achieve a good concordance. I recommend the paper to publication as it is.
Author Response
We are grateful to the reviewer
We have revised the English language in sections: abstract, introduction, discussion.